# Efficacy and safety of pembrolizumab in cervical cancer: Protocol for systematic review and meta-analysis of randomized clinical trials

Dary Medeiros Dantas[1], Amaxsell Thiago Barros de Souza[2], Juliana Dantas de Araújo Santos Camargo[3], Ana Paula Ferreira Costa[3,4], Ayane Cristine Alves Samento[3,4], Andrea Juliana Pereira de Santana Gomes[5], Eduardo Pereira de Azevedo[6], Kleyton Santos de Medeiros[3,4], Isis Kelly dos Santos[7], Ricardo Ney Cobucci[1,6]*

1 Graduate Program in Sciences Applied to Women's Health, Federal University of Rio Grande do Norte, Natal, Rio Grande do Norte, Brazil, 2 Department of Pharmacy, Federal University of Rio Grande do Norte, Natal, Rio Grande do Norte, Brazil, 3 Graduate Program in Health Science, Federal University of Rio Grande do Norte, Natal, Rio Grande do Norte, Brazil, 4 League Against Cancer, Research and Innovation Teaching Institute, Natal, Rio Grande do Norte, Brazil, 5 Department of Oncology, League Against Cancer, Natal, Rio Grande do Norte, Brazil, 6 Graduate Program in Biotechnology, Potiguar University, Natal, Rio Grande do Norte, Brazil, 7 Department of Physical Education, Federal University of Rio Grande do Norte, Natal, Rio Grande do Norte, Brazil

* ricardo.cobucci.737@ufrn.edu.br

## Abstract

### Purpose

This paper reports a systematic review and meta-analysis protocol that will be used to evaluate the efficacy and safety of pembrolizumab, alone or combined with bevacizumab and other therapies, in adult women with cervical carcinoma from stage IB2 onwards.

### Methods

The protocol follows PRISMA-P recommendations and was registered on PROSPERO (CRD42024531233). The search will be conducted without restrictions on language and year of publication in the following databases: Pubmed, Embase, Scopus, Web of Science, Cancerlit, The World Health Organization (WHO), International Clinical Trials Registry Platform (ICTRP) and Clinical Trials Registry Platform. Grey literature will be searched using the following sources: Clinicaltrials.gov, Google Scholar and Opengrey. Manual search will be carried out for the reference lists of eligible studies. The studies will be selected independently by two reviewers and all completed or ongoing randomized clinical trials that evaluated the efficacy and safety of pembrolizumab, used alone or combined with chemotherapy, radiotherapy, bevacizumab or surgery, in adult women diagnosed with cervical cancer, will be included. The data extraction will include population characteristics, type of treatment and main outcomes of studies. The methodological quality of the studies will be assessed using the Cochrane Risk of Bias 2.0. The certainty of the evidence will be rated using the Grading of Recommendations, Assessment, Development, and Evaluations (GRADE).

**Data Availability Statement:** Deidentified research data will be made publicly available when the study is completed and published.

**Funding:** CAPES- Fundação Coordenação de Aperfeiçoamento de Pessoal de Nível Superior.

**Competing interests:** The authors have declared that no competing interests exist.

## Conclusions

The findings will be presented in narrative summary tables and a quantitative synthesis will be conducted using the 'meta' package of R software, version 4.3.1. This future systematic review may contribute with quality evidence for clinical decision-making on the use of pembrolizumab in women with cervical cancer.

## Introduction

The incidence and mortality of cervical cancer are still high. In 2020, 604 thousand new cases and 342 thousand deaths were reported worldwide [1]. However, prevention, early detection and treatment of cervical cancer (CC), which is one of the main causes of death among women, has improved in developed countries due to better organization of screening and as a result of the effectiveness of HPV vaccine and therapeutic advances [2].

Currently, the type of treatment of CC is dictated by the staging of the disease. The main treatment modalities include surgery, radiotherapy and chemotherapy. Standard first-line therapy for persistent, recurrent, or metastatic cervical cancer is platinum- and paclitaxel-based chemotherapy associated with bevacizumab [3, 4].

However, evidence suggests that synergism between chemotherapy and immunomodulators can increase antitumor activity, resulting in greater survival and reduced mortality [5]. Immunomodulators have the potential to modulate the tumor microenvironment, stimulating immune system cells to initiate an acute inflammatory reaction that leads to the destruction of malignant tissue [6]. The PD-1 receptor-ligand interaction is the main pathway used by tumors to suppress the T cell-mediated adaptive immune response [7, 8]. Pembrolizumab, an anti-PD-1 monoclonal antibody, blocks the interaction between PD-1 and its ligand PD-L1, suppressing T cell apoptosis, therefore contributing to the proliferation of these cells that will prevent tumor growth [9, 10]. Frenel et al. showed that pembrolizumab was efficacious with very low toxicity, which might indicate a promising drug as a monotherapy or to be combined with standard treatment in women with advanced, recurrent or persistent cervical cancer [11].

However, in many countries, pembrolizumab is not recommended in official protocols as an isolated therapy or combined with chemotherapy and bevacizumab. Likewise, it is not indicated as first-line treatments for persistent, recurrent and metastatic cervical carcinoma. In a recent meta-analysis with seven clinical trials and 727 women, the authors concluded that pembrolizumab increases survival, but highlighted the need for more well-conducted clinical trials to confirm these findings [12]. In a systematic review of 51 studies that evaluated the efficacy and safety of immunotherapy in cervical cancer, the authors highlighted that pembrolizumab is an effective drug, but additional studies are necessary [13]. In addition, recent trials have suggested that combining bevacizumab with pembrolizumab can improve overall survival for patients with persistent, recurrent, or metastatic cervical cancer [14, 15]. However, the evidence on the use of pembrolizumab with bevacizumab in cases of recurrent and advanced cervical cancer needs to be combined in a meta-analysis so that its use can be incorporated into clinical guidelines.

New randomized clinical trials (RCTs) involving women with advanced stages of cervical carcinoma have been performed with the purpose of evaluating the efficacy and safety of pembrolizumab. The results of these studies have been published and are ongoing [16]. Therefore, the aim of this manuscript is to present this protocol that proposes to combine the results of these RCTs in which pembrolizumab was evaluated, either as an isolated or combined

treatment, in women with cervical carcinoma. A systematic review of randomized clinical trials can generate new, higher quality evidence on increased survival and reduced mortality as a result of the treatment with pembrolizumab, which will help to assist in clinical decision-making as well as to include this drug in cervical cancer management guidelines.

## Methods

This protocol follows the Preferred Reporting Items for Systematic Reviews and Meta-Analysis Protocol (PRISMA-P) guideline and was registered in the International Prospective Register of Systematic Reviews (PROSPERO/CRD42024531233) [17].

The systematic review will be conducted and reported in accordance with the PRISMA checklist, which will be updated whenever necessary to ensure proper rigor and transparency of this review [18]. As our research will involve the synthesis of existing primary data, it does not require ethical approval.

### Research question

Is pembrolizumab, used alone or associated with bevacizumab, surgery, radiotherapy, and chemotherapy, as effective and safe as the standard treatments for recurrent and metastatic cervical cancer?

### Search strategy

Study investigators and a medical librarian collaboratively developed the search strategy to identify primary studies that examined whether pembrolizumab is effective and safe, either alone or combined with a standard treatment, in the management of cervical cancer.

Eight databases will be searched from inception until July 2024: Pubmed, Embase, Scopus, Web of Science, Cancerlit, The World Health Organization (WHO), International Clinical Trials Registry Platform (ICTRP), Clinical Trials Registry Platform and Cochrane Library. Grey literature will be searched using the following sources: Clinicaltrials.gov, Google Scholar and Opengrey. Additionally, there will be a search for abstracts of randomized clinical trials presented in major international congresses in the last five years. The reference list of included articles and relevant reviews identified through the search will be scanned to determine any potential article that is relevant to this review.

The search strategy will be designed with a combination of MESH/EMTREE terms and synonyms, being adapted for each database. There will be no restrictions regarding publication date and language.

The Pubmed search strategy will be developed in collaboration among all authors and a health sciences librarian with expertise in systematic review searching. Subsequently, the search strategy will be peer-reviewed by an external health sciences librarian in accordance with the PRESS standard [19]. After acceptance of the Pubmed search strategy, it will be translated to the other databases. The draft of the Pubmed search strategy is as follows: ("Pembrolizumab" OR "SCH-900475" OR "lambrolizumab" OR "MK-3475" OR "Keytruda") OR ("Bevacizumab" OR "Mvasi" OR "Bevacizumab-awwb" OR "Avastin") AND ("surgery" OR "radiotherapy OR "chemotherapy") AND ("uterine cervical neoplasms" OR "cervical neoplasm" OR "cancer of the uterine cervix" OR "cervical cancer" OR "uterine cervical cancer" OR "cervix cancer") AND ("randomized clinical trial" OR "controlled clinical trial" OR "Trials, Randomized Clinical" OR "intervention study" OR "clinical study" OR "clinical studies").

After validation of the proposed search strategy for PubMed by the librarian, an adaptation of this strategy was used for EMBASE as follows: ('Pembrolizumab' OR 'SCH-900475' OR 'lambrolizumab' OR 'MK-3475' OR 'Keytruda') OR ('Bevacizumab' OR 'Mvasi' OR

'Bevacizumab-awwb' OR 'Avastin') AND ('surgery'/exp OR 'radiotherapy'/exp OR 'chemotherapy'/exp) AND ('uterine cervical neoplasms'/exp OR 'cervical neoplasm'/exp OR 'cancer of the uterine cervix'/exp OR 'cervical cancer'/exp OR 'uterine cervical cancer'/exp OR 'cervix cancer'/exp) AND ('randomized controlled trial'/exp OR 'controlled clinical trial'/exp OR 'clinical trial'/exp OR 'intervention study'/exp OR 'clinical study'/exp).

## Eligibility criteria

In the systematic review, studies that meet the following criteria will be included: randomized clinical trials, which evaluated the efficacy and safety of pembrolizumab, used alone or combined with chemotherapy, radiotherapy, bevacizumab or surgery, in all adult women diagnosed with cervical cancer that received pembrolizumab treatment, from IB2 tumoral stage according to the classification of the International Federation of Gynecology and Obstetrics (FIGO) [20]. There will be no restrictions regarding the follow-up time and setting of the studies.

Only original studies will be eligible for inclusion in this study, including completed and ongoing clinical trials with preliminary results. Studies that involve mixed populations (e.g., cervical cancer patients along with other cancer types), case reports, preprint, narrative and systematic reviews and observational studies will be excluded.

Studies will be selected based on the following eligibility criteria:

P(Population): Adult women diagnosed with advanced and high-risk stage IB2 cervical cancer onwards who had been diagnosed and previously treated, or not, with pembrolizumab alone, or in combination with bevacizumab, surgery, radiotherapy, and chemotherapy.

I(Intervention): Pembrolizumab used either alone or combined with Bevacizumab, chemotherapy, surgery and other therapies.

C(Comparison): Surgery, radiotherapy, platinum-based chemotherapy, bevacizumab and placebo.

O(Outcomes): Mortality rate as primary outcome. Overall survival (OS), disease-free survival (DFS), complete remission (CR), partial response (PR), stable disease (SD), disease progression (DP), overall response rate (ORR), disease control rate (DCR), median progression free survival (mPFS), median OS (mOS), and adverse events associated with the use of pembrolizumab will be considered as secondary outcomes.

S (Study design): Randomized clinical trials.

## Definition of measures

Mortality rate: number of deaths, with cancer as the main cause of death, in a population affected by cervical carcinoma.

Overall survival: time interval between the diagnosis of cervical carcinoma and death from any cause.

Disease-free survival: period of time after the end of primary treatment for cervical cancer, during which the patient survived without any signs or symptoms of this cancer.

Complete remission: all signs and symptoms of cancer have disappeared, although cancer still may be in the patient's body.

Partial response: decrease in the tumor size, or in the extent of cancer in response to treatment.

Stable disease: when the disease is neither increasing nor decreasing in extent or severity.

Disease progression: increase in the sum of maximum tumor diameters of at least 20%, together with the development of any new lesions or an unequivocal increase in non-measurable malignant disease.

Overall response rate: percentage of patients whose cancer shrinks or disappears after treatment.

Disease control rate: percentage of patients with advanced cancer whose therapy has led to a complete or partial response or stable disease.

Progression free survival: length of time during and after the treatment of the cancer in which the patient lives with the disease without getting worse.

Adverse events: any abnormal clinical finding temporarily associated with the use of therapy for cervical carcinoma, according to CTCAE V5 [21].

## Data collection and analysis

The retrieved studies will be exported to Rayyan software (Mourad Ouzzani, University of Oxford, UK). All titles and abstracts will be screened independently by two reviewers (DMD and ATBS). Disagreements will be resolved through discussion with a third reviewer (ACAS). To ensure uniform interpretation of the eligibility criteria, a pilot screening of 200 articles from the Pubmed search will be performed before the initiation of the title and abstract screening phase.

After selecting the studies by titles and abstracts, three independent reviewers (DMD, ATBS, ACAS) will read the articles in their entirety and select the studies according to the eligibility criteria, with disagreements being resolved by consensus with the participation of a fourth reviewer (RNC). After this selection, they will independently extract data from eligible studies using a predefined extraction scheme.

The data from each study will be extracted for the following categories: author, year of publication, country, categories of participants, sample size, study design, histological type, cancer staging, type of intervention, follow-up period, mortality rate, overall survival, disease-free survival, adverse events, and main findings. If additional details are required, the authors of the study will be contacted to provide all necessary unreported data. The data not provided from those studies in which three attempts were made to contact the authors by email, will be excluded from the meta-analysis, with information from these studies remaining only in the narrative synthesis of the systematic review.

The PRISMA 2020 flow diagram (Fig 1) will be used to present the study identification, screening and inclusion.

## Bias risk assessment

The bias risk of the selected articles will be assessed independently by two reviewers using the updated Cochrane risk-of-bias tool for randomized trials (RoB2) [22]. The specific criteria that will be evaluated include random sequence generation to assess selection bias, allocation concealment to further evaluate selection bias, blinding of participants and personnel to address performance bias, analysis of incomplete outcome data to determine attrition bias, examination of selective reporting to identify reporting bias, and the identification of any other potential sources of bias. In case of insufficient details reported in the study, authors will be contacted. Discrepancies in regard to the assessment of risk of bias will be solved through discussion with a third author (RNC).

Randomized clinical trials retrieved from the grey literature will have their risk of bias rigorously assessed by this tool. In cases of studies with incomplete data that compromise the

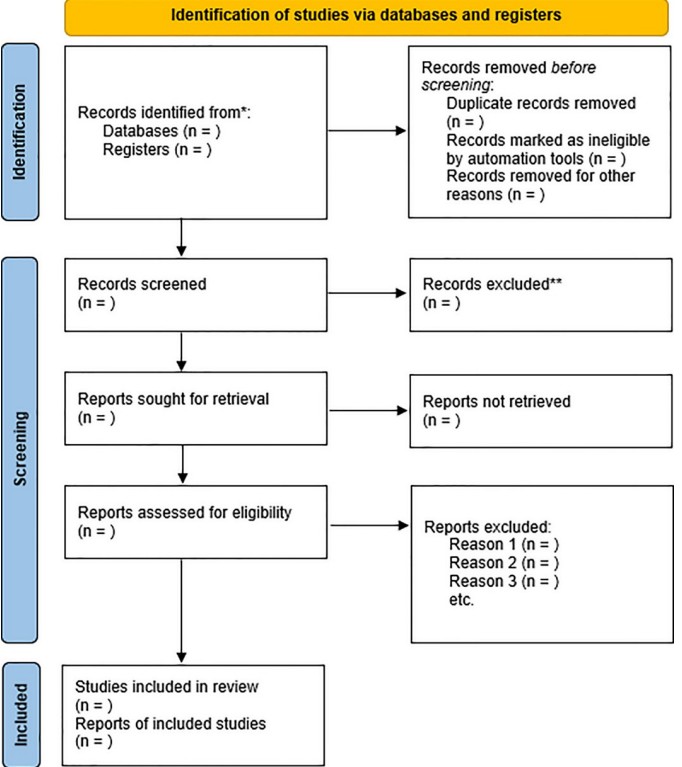

**Fig 1. PRISMA flow diagram for systematic review and meta-analysis.**

rigorous assessment of the risk of bias, three attempts will be made to contact the authors via email, and only the RCTs whose data were recovered will be included in the meta-analysis.

Whenever possible, funnel plots will be used to assess the potential existence of publication bias, complemented by Egger's weighted correlation and Begg's regression intercept at a 5% significance level [23].

## Certainty of evidence

The Grading of Recommendations, Assessment, Development and Evaluation (GRADE) framework will be used to grade the quality of evidence for each intervention as high, moderate, low or very low [24]. Overall, upgrading or downgrading the quality of evidence depends on the risk of bias, imprecision, inconsistency, and indirectness of the findings. Any disagreement regarding the assessment of certainty of evidence will be resolved through consensus among the authors, which may guarantee that the conclusions provide a balanced view of the available evidence.

Summary of Findings (SoF) table will be created and will present the following information for primary and secondary outcomes: absolute risks for the treatment and control, estimates of Odds Ratio (OR), Hazard Ratio (HR), Relative Risk (RR), or Incidence Rate Ratio (IRR), directness, heterogeneity, precision, and risk of publication bias of the results.

All evidence graded as low or very low certainty will be considered with careful caution and will be specifically highlighted in the conclusion of the future paper. This will guide the clinical decision-making regarding the efficacy and safety of pembrolizumab, emphasizing that lower quality evidence should be interpreted more carefully in clinical recommendations.

## Statistical analysis

A quantitative synthesis will be conducted using the 'meta' package of R software, version 4.3.1. For dichotomous outcomes, the Odds Ratio (OR), Hazard Ratio (HR), Relative Risk (RR), or Incidence Rate Ratio (IRR), along with their 95% Confidence Intervals (CI 95%), will be extracted or calculated for each study. The Mean Difference (MD) or Standardized Mean Difference (SMD) will be computed for continuous data.

In cases of high heterogeneity ($I^2 \geq 50\%$), the random-effects model will be employed to combine the studies and to calculate the OR and CI 95% using the DerSimonian-Laird algorithm. We will use random-effects models to analyze survival data. In cases of missing data, we will contact the authors of the studies to obtain the missing information for more accurate inference. Additionally, we will conduct a sensitivity analysis to assess the impact of the missing data on our results.

In cases where the $I^2$ statistic is substantially high, we will discuss its potential implications for the overall interpretation of the results. Additionally, only clinical trials whose data are recovered after requesting from the primary study authors will be considered for quantitative synthesis, due to the fact the literature does not recommend statistically estimating unavailable data as this can compromise the quality of the meta-analysis results.

In the case of a few clinical trials with high heterogeneity and very high risk of bias, a meta-analysis will not be conducted, and the results of these studies will be presented in a qualitative synthesis. Synthesis Without Meta-analysis (SWiM) guidelines will be used when preparing and conducting the narrative synthesis whenever data are insufficient to conduct meta-analysis and to estimate an effect [25].

## Subgroup analysis

Subgroup analysis will be conducted by separating the doses of pembrolizumab and combining groups that received the same doses of just this drug, or those that received combination therapy with bevacizumab, with the aim of verifying if there are differences in primary and secondary outcomes. Additionally, analysis will be performed according to the follow-up time of the clinical trials and the way adverse effects were assessed.

Analysis by age group of the women and by the stage of the cervical cancer, as well as by classifying the cancer as recurrent or metastatic and by its histological type (squamous, non-squamous) will be assessed by comparing outcomes after treatment with pembrolizumab alone or in combination according to the stage/type of cancer.

Finally, the groups will also be divided according to the expression levels of immune checkpoint molecules, such as PD-1 and PD-L1, with the purpose of assessing whether certain molecular signatures are associated with the efficacy of pembrolizumab treatment.

## Sensitivity analysis

The robustness of the meta-analysis findings will be tested through sensitivity analysis, employing a one-by-one elimination method to pinpoint the sources of heterogeneity. Sensitivity analysis will also be conducted for testing the impact of the inclusion of low-quality studies.

Additionally, sensitivity will be tested by combining clinical trials with the same follow-up time, the same method of adverse effects assessment, and standardized outcome measures, increasing the ability to identify potential causes of heterogeneity and to deliver more reliable quantitative results.

### Missing data

In instances where data of interest are missing or unclear, the research team will proceed to contact the corresponding author via email to obtain the necessary information. If these efforts are unsuccessful after 3 attempts, the data will be excluded from the quantitative analysis. This limitation will be acknowledged and discussed in the discussion section.

## Discussion

Despite constant advances in cancer therapy, high-risk, advanced, recurrent, or metastasized cervical cancers still carry a poor prognosis, with recurrence occurring in most patients within 2 years [26]. A deep understanding of tumor immunity has enabled immunotherapy to become one of the most promising approaches to treat cancers. Although chemotherapy remains the standard of care for recurrent or metastatic cervical cancer, treatment options remain limited for patients experiencing tumor progression post-chemotherapy. Although pembrolizumab has already been approved by the US Food and Drug Administration (FDA), clinical trials that have addressed only the efficacy and safety of this drug as an isolated therapy or combined with standard treatment for cervical cancer present controversial results. Thus, better quality evidence is needed to incorporate pembrolizumab into the guidelines of cervical cancer treatment [27–30].

Due to the high mortality and low survival rates of women affected by advanced, metastatic and recurrent cervical tumors, even when treated with the currently available therapies, new evidence on therapies capable of contributing to the reduction of deaths and increasing the overall and free survival of this disease can impact the epidemiology of cervical carcinoma [26].

New studies with high methodological rigor and higher levels of evidence may allow healthcare professionals to base themselves on the results of these RCT meta-analyses to recommend immunomodulators and other drugs, whose efficacy and safety have been proven for the treatment of cervical cancer [31].

Recently, immunobiologicals (pembrolizumab, cemiplimab and others) have emerged as therapeutic options that could provide durable responses with a significant impact on the overall survival and mortality rates. Several trials in monotherapy or in combination with chemotherapy or with bevacizumab have shown very promising results. Tewari et al. reported that the inclusion of pembrolizumab to chemotherapy with or without bevacizumab improved the overall survival by subgroups of patients with persistent, recurrent, or metastatic cervical cancer [14]. However, there is still a great need for higher quality studies capable of determining more precisely which patients must receive the greatest benefit from all of these mentioned drugs, but also to identify patients with specific molecular characteristics who could benefit from other targeted therapies [15, 27, 31].

Therefore, the proposal for a systematic review presented herein of randomized clinical trials that evaluated the efficacy and safety of pembrolizumab, either as monotherapy or combined with established treatments for cervical cancer, was based on the need for high-level evidence that supports clinical decision-making by healthcare professionals who might consider adopting this drug as an option to increase survival and reduce mortality of patients with cervical cancer. However, this future review may present as limitations the small number of randomized clinical trials with pembrolizumab, which might result in methodological problems such as small sample size, high risk of bias and very low certainty of evidence. The publication of the protocol, after careful peer review that ensures a judicious methodology, aims to guarantee the quality of the systematic review.

## Conclusion

The role of immunotherapy in the treatment of cervical cancer is increasing, and specific biomarkers will also need to be explored, such as PD-1 expression, microsatellite instability, and mismatch repair deficiency. Several questions remain, including the exact contribution of immunotherapy to survival and mortality in advanced cervical cancer. The use of pembrolizumab still faces numerous challenges and open questions, such as determining the most appropriate starting time, addressing inherent and emerging resistance, and identifying the ideal cohort of patients for this therapy. Thus, gaps might be clarified in a future systematic review.

## Supporting information

**S1 Checklist. PRISMA-P (Preferred Reporting Items for Systematic review and Meta-Analysis Protocols) 2015 checklist: Recommended items to address in a systematic review protocol*.**
(DOCX)

## Author Contributions

**Conceptualization:** Dary Medeiros Dantas, Amaxsell Thiago Barros de Souza, Ana Paula Ferreira Costa, Ayane Cristine Alves Samento, Kleyton Santos de Medeiros, Isis Kelly dos Santos, Ricardo Ney Cobucci.

**Data curation:** Juliana Dantas de Araújo Santos Camargo.

**Formal analysis:** Amaxsell Thiago Barros de Souza.

**Funding acquisition:** Ricardo Ney Cobucci.

**Investigation:** Dary Medeiros Dantas, Amaxsell Thiago Barros de Souza, Ana Paula Ferreira Costa, Ayane Cristine Alves Samento, Andrea Juliana Pereira de Santana Gomes.

**Methodology:** Dary Medeiros Dantas, Amaxsell Thiago Barros de Souza, Juliana Dantas de Araújo Santos Camargo, Ana Paula Ferreira Costa, Ayane Cristine Alves Samento, Kleyton Santos de Medeiros, Isis Kelly dos Santos, Ricardo Ney Cobucci.

**Project administration:** Isis Kelly dos Santos, Ricardo Ney Cobucci.

**Software:** Juliana Dantas de Araújo Santos Camargo, Ricardo Ney Cobucci.

**Supervision:** Andrea Juliana Pereira de Santana Gomes, Eduardo Pereira de Azevedo, Kleyton Santos de Medeiros, Isis Kelly dos Santos, Ricardo Ney Cobucci.

**Validation:** Andrea Juliana Pereira de Santana Gomes, Eduardo Pereira de Azevedo, Kleyton Santos de Medeiros, Isis Kelly dos Santos, Ricardo Ney Cobucci.

**Visualization:** Ayane Cristine Alves Samento.

**Writing – original draft:** Dary Medeiros Dantas, Amaxsell Thiago Barros de Souza, Ana Paula Ferreira Costa, Ayane Cristine Alves Samento.

**Writing – review & editing:** Andrea Juliana Pereira de Santana Gomes, Eduardo Pereira de Azevedo, Kleyton Santos de Medeiros, Isis Kelly dos Santos, Ricardo Ney Cobucci.

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
