## [Decision Letter · Decision Letter 0]

1 Sep 2024

PONE-D-24-34168Efficacy and safety of Pembrolizumab in cervical cancer: protocol for systematic review and meta-analysis of randomized clinical trialsPLOS ONE

Dear Dr. Cobucci,

Thank you for submitting your manuscript to PLOS ONE. After careful consideration, we feel that it has merit but does not fully meet PLOS ONE’s publication criteria as it currently stands. Therefore, we invite you to submit a revised version of the manuscript that addresses the points raised during the review process.

We look forward to receiving your revised manuscript.

Kind regards,

Cho-Hao Howard Lee, M.D.

Academic Editor

PLOS ONE

“CAPES- Fundação Coordenação de Aperfeiçoamento de Pessoal de Nível Superior”

Reviewers' comments:

Reviewer's Responses to Questions

**Comments to the Author**

1. Does the manuscript provide a valid rationale for the proposed study, with clearly identified and justified research questions?

Reviewer #1: Partly

Reviewer #2: Yes

Reviewer #3: Yes

Reviewer #4: Partly

Reviewer #5: Yes

2. Is the protocol technically sound and planned in a manner that will lead to a meaningful outcome and allow testing the stated hypotheses?

Reviewer #1: Partly

Reviewer #2: Yes

Reviewer #3: Partly

Reviewer #4: Partly

Reviewer #5: Yes

3. Is the methodology feasible and described in sufficient detail to allow the work to be replicable?

Reviewer #1: Yes

Reviewer #2: Yes

Reviewer #3: Yes

Reviewer #4: Yes

Reviewer #5: No

4. Have the authors described where all data underlying the findings will be made available when the study is complete?

Reviewer #1: Yes

Reviewer #2: Yes

Reviewer #3: Yes

Reviewer #4: Yes

Reviewer #5: Yes

5. Is the manuscript presented in an intelligible fashion and written in standard English?

Reviewer #1: Yes

Reviewer #2: Yes

Reviewer #3: Yes

Reviewer #4: Yes

Reviewer #5: Yes

6. Review Comments to the Author

You may also provide optional suggestions and comments to authors that they might find helpful in planning their study.

Reviewer #1: Overview:

This protocol for a systematic review and meta-analysis on the efficacy and safety of pembrolizumab in cervical cancer is timely and addresses a critical area in cancer immunotherapy. The study follows a rigorous methodology with adherence to PRISMA-P guidelines and a clear research focus. However, there are several areas where the protocol needs further elaboration and clarification before it can be considered for publication.

Strengths:

Relevant Research Topic: The use of pembrolizumab in cervical cancer is a highly relevant and important topic given the increasing use of immunotherapy in oncology. This protocol addresses a significant knowledge gap in understanding the efficacy and safety of pembrolizumab, particularly in women with advanced-stage cervical cancer.

Methodological Rigor: The use of PRISMA-P recommendations, PROSPERO registration, and detailed plans for data extraction and synthesis demonstrates a solid methodological framework. The authors have clearly outlined their search strategy, inclusion criteria, and statistical approach, which enhances the reproducibility and reliability of the systematic review.

Detailed Risk of Bias Assessment: The authors have indicated their intention to use the Cochrane Risk of Bias 2.0 tool, which is a strength. The explicit focus on minimizing selection, performance, and reporting bias shows the team’s attention to producing high-quality evidence.

Major Areas for Revision:

Lack of Justification for Pembrolizumab Dosages and Treatment Variability:

The manuscript mentions that subgroup analyses will be conducted based on different doses of pembrolizumab and its combination with other therapies, but there is no clear explanation of how the dosage variability will be categorized or analyzed. Since pembrolizumab’s effects may depend on dosage and treatment combinations, I strongly recommend that the authors provide more detailed justification on how they plan to handle this heterogeneity in the analysis.

Insufficient Clarification on Study Timeframes:

The timeframe for inclusion of randomized clinical trials is not explicitly stated. Although the search strategy mentions that the databases will be searched up until July 2024, it is unclear whether the authors are only including trials that have been completed or also ongoing trials with preliminary results. Clarifying this point would help set expectations regarding the evidence that will be synthesized.

Ambiguity in Handling Grey Literature:

The protocol indicates that grey literature will be included, but there is limited information about how the authors plan to assess the quality of non-peer-reviewed studies. Given that grey literature can introduce higher risk of bias, it is critical that the authors provide a detailed plan for evaluating these sources using robust criteria. Without a clear strategy, the inclusion of grey literature could compromise the reliability of the findings.

Statistical Analysis Plan Lacks Detail:

While the protocol mentions using the ‘meta’ package in R for statistical analysis, it does not elaborate on the specific models or tests that will be used for certain key outcomes. For example, will the authors use fixed-effects or random-effects models for survival analysis, and how will they handle studies with missing data? More detail on how these challenges will be addressed is needed for transparency and reproducibility.

Unclear Plan for Addressing Heterogeneity:

The authors acknowledge that heterogeneity is likely, especially considering the variability in study design, pembrolizumab doses, and combined therapies. However, the protocol lacks a clear plan for handling this heterogeneity. I recommend a more detailed strategy for conducting subgroup and sensitivity analyses to identify and address sources of heterogeneity.

Limited Focus on Patient Populations:

The protocol states that women with cervical cancer at stage IB2 onwards will be included. However, more clarity is needed on how the authors will handle studies that involve mixed populations (e.g., cervical cancer patients alongside other cancer types). Will such studies be excluded, or will the cervical cancer cohort be analyzed separately? This needs to be addressed to avoid confusion and ensure that the meta-analysis is focused on cervical cancer.

Specific Suggestions

Expand on Search Strategy Development:

Although the search strategy for PubMed is briefly outlined, it would be beneficial to provide a more detailed breakdown of how search terms were chosen and how the strategy will be adapted for other databases. Providing an example of search terms for another database (e.g., Embase or Scopus) would help readers understand the thoroughness of the search process.

Provide a Clearer Outline of the Subgroup Analyses:

While subgroup analyses are mentioned, more detail is needed on how these will be conducted. For instance, how will the authors define “combined therapies,” and what specific interactions between pembrolizumab and other drugs (e.g., bevacizumab) will be evaluated? Clarifying this would improve the methodological transparency of the protocol.

Clarify the Role of the GRADE Framework:

The protocol briefly mentions that the GRADE framework will be used to assess the certainty of evidence, but it does not explain how this assessment will influence the interpretation of findings. Will low-certainty evidence be weighted less heavily in the overall conclusions? Providing more detail on how GRADE will be applied would be helpful for understanding the implications of the findings.

Consider Including More Detailed Sensitivity Analyses:

The sensitivity analyses described in the protocol are somewhat limited in scope. I suggest expanding this section to include other factors that may affect study outcomes, such as varying definitions of adverse events or different follow-up periods across trials. A more thorough sensitivity analysis would improve the robustness of the findings.

Conclusion:

This protocol has the potential to contribute valuable insights into the efficacy and safety of pembrolizumab in cervical cancer. However, several critical areas need revision, including clarification of subgroup analyses, handling of grey literature, and expansion of the statistical analysis plan. With these revisions, the protocol will be better positioned to generate high-quality, reliable evidence that can guide clinical decision-making. Therefore, I recommend major revisions to strengthen the methodology and transparency of this work.

Reviewer #2: This study proposed a protocol for systematic review and meta-analysis for Pembrolizumab in cervical cancer. It is a through study protocol with strict inclusion criteria and rigorous data collection plan, and reasonable scrutinization regarding data analysis. However, there are some issues to be addressed before publication:

1. A major concern is the inclusion criteria. The authors intend to include previously untreated cervical cancer patients, however, currently, immunotherapy is usually used in late stage or recurrent cases as a salvage strategy. Only including previously untreated cases might appear somewhat biased and does not reflect actual clinical practice. I would suggest broadening the inclusion criteria to all cervical cancer patients received pembrolizumab treatment.

Other PD-1 inhibitors, such as Nivolumab should be included as well if there are available studies.

2. In search strategy, (lines #129-136) the authors included pembrolizumab and bevacizumab, but left out other standard treatment strategies, such as chemotherapy, radiotherapy, or surgery.

3. Line #152, I don’t think the patients have to be “recently” diagnosed and suggest “recently” be strike out.

4. Line #158, I think bevacizumab should be included as well.

5. For the subgroup analysis, I suggest look at subgroups divided by expression levels of immune checkpoint molecules, such as PD-1, PD-L1, to see if certain molecular signatures are associated with ICI treatment effects. This will shed lights on what patient populations will benefit most from ICIs.

5. Line #184, “DMB” should be “DMD”.

6. Line #45, “will be included” should be deleted for a correct sentence structure.

Reviewer #3: Dear Dr.Ricardo Ney Oliveira Cobucci,

I hope this message finds you well. I have had the opportunity to review your manuscript entitled "Efficacy and safety of Pembrolizumab in cervical cancer: protocol for systematic review and meta-analysis of randomized clinical trials," and I appreciate the efforts you have invested in this significant area of research. While the topic is undeniably important and timely, there are several aspects of the study design and scope that, if addressed, could enhance the manuscript’s contribution to the field significantly.

1. Comprehensive Literature Search: Your protocol outlines a focused search strategy that, while robust, may benefit from an expanded scope to include grey literature and databases such as The World Health Organization (WHO) International Clinical Trials Registry Platform (ICTRP) search portal (http://apps.who.int/trialsearch); Clinical Trials Registry Platform (http://www.who.int/ictrp/en/); European Medicines Agency (EMA)(https://www.ema.europa.eu/ema/); US Food and Drug Administration (FDA) (www.fda.gov). To ensure even more comprehensive coverage, please consider meticulously scanning the abstracts presented at major international congresses during the three years preceding our search. This strategy aims to capture any studies that have been presented at these events but have yet to be fully published. This broader approach will help capture emerging studies and ensure a more comprehensive analysis, enhancing the relevance and applicability of your findings.

2. Search Terms and Inclusion Criteria: The manuscript currently limits its search terms primarily to specific names associated with Bevacizumab. Considering the diversity of treatment modalities in cervical cancer, including chemotherapy, radiotherapy, and surgery, broadening the search terms to encompass these areas could provide a more holistic view of the landscape and potential interactions or comparative efficacies.

3. Discussion of Combined Therapies: Your manuscript would benefit from a more detailed discussion on the role and implications of combining Pembrolizumab with Bevacizumab. Highlighting this in both the introduction and discussion sections would clarify the therapeutic context and relevance of your findings to current treatment protocols.

4. Analysis of Adverse Events: Considering serious adverse events as a secondary endpoint is a prudent choice.

However, a comprehensive analysis of all adverse events, if feasible, would provide a deeper understanding of safety profiles, which is crucial for clinical decision-making.

5. Handling of Missing Data: It is essential to outline a clear and rigorous method for handling missing data in your analysis. This includes efforts to retrieve missing information and statistically estimating unavailable data, which would strengthen the validity and reliability of your study results.

6. Clarity of Research Questions: Finally, I recommend structuring a separate section that distinctly lists the research questions your study seeks to answer, including benefits, risks, and subgroup analyses. This would greatly enhance the clarity and focus of your manuscript, making it more accessible to readers and reviewers.

While the manuscript presents a valuable protocol, addressing these points could significantly strengthen your study's impact and scientific contribution. I believe these enhancements will better position your manuscript for publication, offering a substantial advancement in our understanding of Pembrolizumab's role in treating cervical cancer.

Reviewer #4: The author expressed the need for further research on pembrolizumab as a treatment option for cervical cancer.

However, the outcome selection seems to be not ideal.

Previous meta-analysis used the outcome of complete response (CR), partial response (PR), stable disease (SD), disease progression (PD), overall response rate (ORR), disease control rate (DCR), median progression-free survival (mPFS), and median overall survival (mOS)

The latest RCT reported OS and PFS.

Authors need to re-consider the outcome that would be included.

Reviewer #5: The proposed systematic review and meta-analysis protocol is well-structured, comprehensive, and adheres to established guidelines. The topic is highly relevant given the increasing role of immunotherapy in treating cervical cancer, and the need for high-level evidence to support its use. However, there are several areas where the protocol could be strengthened or clarified to ensure the robustness and reliability of the final review.

1. The research question is clearly stated and addresses a significant gap in the current literature. However, the objectives could benefit from a more explicit statement on how the results of this review will impact clinical guidelines and practice. Consider expanding on the potential implications of the findings.

2. Under “Search Strategy”, while the search terms for PubMed are provided, it would be helpful to include a more detailed explanation of how these terms will be adapted for other databases.

3. Under introduction, “The results of these studies have been published and are ongoing – interesting.” - The predefined extraction scheme is thorough, but it may be beneficial to specify how you will handle missing data or studies are still ongoing with incomplete reporting.

4. For statistical analysis, the plan to use the random-effects model in cases of high heterogeneity is appropriate. However, you might want to also clarify how you will interpret and present results if significant heterogeneity is detected.

5. Also under statistical analysis, it may be useful to specify the criteria for defining the subgroups, particularly in relation to different doses of pembrolizumab and patient characteristics.

7. PLOS authors have the option to publish the peer review history of their article (what does this mean?). If published, this will include your full peer review and any attached files.

Reviewer #1: **Yes: **Xiaoyi Zhang, MD

Reviewer #2: No

Reviewer #3: **Yes: **Bingyu Li

Reviewer #4: No

Reviewer #5: **Yes: **Yuhang Liu

---

## [Author Response · Author response to Decision Letter 0]

9 Sep 2024

Dear Editor Cho-Hao Howard Lee,

We appreciate the pertinent contributions suggested by the reviewers and we truly believe that the revised manuscript meets all the requests. First, we clarify the Journal requirements:

1. Please ensure that your manuscript meets PLOS ONE’s style requirements, including those for file naming. The PLOS ONE style templates can be found at

“CAPES- Fundação Coordenação de Aperfeiçoamento de Pessoal de Nível Superior”

Please state what role the funders took in the study. If the funders had no role, please state: “The funders had no role in study design, data collection and analysis, decision to publish, or preparation of the manuscript.”

Authors’ responses:

1. The revised manuscript meets PLOS ONE’s style requirements.

2. The correct grant number for the awards received for the study is included in the ‘Funding Information’ section.

3. The correct statement “The funders had no role in study design, data collection and analysis, decision to publish, or preparation of the manuscript.” has been added to the revised manuscript as well as in the cover letter.

4. The reference list has been reviewed. No retracted articles are included, and the new additions have been highlighted in red in the revised manuscript with track changes.

Reviewer #1:

Lack of Justification for Pembrolizumab Dosages and Treatment Variability:

-The manuscript mentions that subgroup analyses will be conducted based on different doses of pembrolizumab and its combination with other therapies, but there is no clear explanation of how the dosage variability will be categorized or analyzed. Since pembrolizumab’s effects may depend on dosage and treatment combinations, I strongly recommend that the authors provide more detailed justification on how they plan to handle this heterogeneity in the analysis.

Authors’ response: We appreciate the observation. The subgroup analysis is more detailed in the revised manuscript, with explanations on how groups will be analyzed concerning the different doses of pembrolizumab and with the separation of studies where the treatment was either alone or combined with other drugs. 

-Insufficient Clarification on Study Timeframes:

The timeframe for inclusion of randomized clinical trials is not explicitly stated. Although the search strategy mentions that the databases will be searched up until July 2024, it is unclear whether the authors are only including trials that have been completed or also ongoing trials with preliminary results. Clarifying this point would help set expectations regarding the evidence that will be synthesized.

Authors’ response: We agree with the reviewer’s suggestion and have modified the inclusion criteria in the abstract and methodology sections of the revised manuscript, making it clear that ongoing and completed randomized clinical trials will be included. 

-Ambiguity in Handling Grey Literature:

The protocol indicates that grey literature will be included, but there is limited information about how the authors plan to assess the quality of non-peer-reviewed studies. Given that grey literature can introduce higher risk of bias, it is critical that the authors provide a detailed plan for evaluating these sources using robust criteria. Without a clear strategy, the inclusion of grey literature could compromise the reliability of the findings.

Authors’ response: The search for studies published in grey literature databases is a recommendation for systematic reviews as an appropriate strategy according to PRISMA. We agree that these studies may add a high risk of bias to the future review, but in the revised manuscript, we have added a text explaining the rigor of the risk of bias assessment for all included RCTs using the Cochrane tool. We also stated that this risk will be highlighted in the evidence certainty assessment table following the GRADE guideline and noted in the limitations in the discussion, as well as in the recommendations of the conclusion of the future review. 

-Statistical Analysis Plan Lacks Detail:

While the protocol mentions using the ‘meta’ package in R for statistical analysis, it does not elaborate on the specific models or tests that will be used for certain key outcomes. For example, will the authors use fixed-effects or random-effects models for survival analysis, and how will they handle studies with missing data? More detail on how these challenges will be addressed is needed for transparency and reproducibility.

Authors’ response: We agree with the recommendations, and the new text on statistical analysis clarifies how survival analysis will be conducted and how studies with missing data will be handled. 

-Unclear Plan for Addressing Heterogeneity:

The authors acknowledge that heterogeneity is likely, especially considering the variability in study design, pembrolizumab doses, and combined therapies. However, the protocol lacks a clear plan for handling this heterogeneity. I recommend a more detailed strategy for conducting subgroup and sensitivity analyses to identify and address sources of heterogeneity.

Authors’ response: A more detailed strategy for conducting subgroup and sensitivity analyses is described in the revised manuscript with the purpose of identifying and addressing sources of heterogeneity. 

-Limited Focus on Patient Populations:

The protocol states that women with cervical cancer at stage IB2 onwards will be included. However, more clarity is needed on how the authors will handle studies that involve mixed populations (e.g., cervical cancer patients alongside other cancer types). Will such studies be excluded, or will the cervical cancer cohort be analyzed separately? This needs to be addressed to avoid confusion and ensure that the meta-analysis is focused on cervical cancer.

Authors’ response: In the methodology section, we stated that studies involving mixed populations (e.g., cervical cancer patients along with other cancer types) will not be included. 

-Expand on Search Strategy Development:

Although the search strategy for PubMed is briefly outlined, it would be beneficial to provide a more detailed breakdown of how search terms were chosen and how the strategy will be adapted for other databases. Providing an example of search terms for another database (e.g., Embase or Scopus) would help readers understand the thoroughness of the search process.

Authors’ response: In the first manuscript, it was explained that the strategy would be shared with a librarian who would test the PubMed draft, included in the methodology, and after validation, adapt this strategy for the other databases, following a recommendation from the literature. However, we have considered the reviewer’s suggestion and presented in the methodology of the revised manuscript the draft of the strategy that will be tested in EMBASE. 

-Provide a Clearer Outline of the Subgroup Analyses:

While subgroup analyses are mentioned, more detail is needed on how these will be conducted. For instance, how will the authors define “combined therapies,” and what specific interactions between pembrolizumab and other drugs (e.g., bevacizumab) will be evaluated? Clarifying this would improve the methodological transparency of the protocol.

Authors’ response: We appreciate the suggestion. The subgroup analysis has been improved to clarify how the analysis will be separated according to whether the groups received monotherapy or combined therapy, as well as defining the analysis by age group, histological type, and follow-up time. 

-Clarify the Role of the GRADE Framework:

The protocol briefly mentions that the GRADE framework will be used to assess the certainty of evidence, but it does not explain how this assessment will influence the interpretation of findings. Will low-certainty evidence be weighted less heavily in the overall conclusions? Providing more detail on how GRADE will be applied would be helpful for understanding the implications of the findings.

Authors’ response: Relevant suggestion and more details on how GRADE will be applied have been added. 

-Consider Including More Detailed Sensitivity Analyses:

The sensitivity analyses described in the protocol are somewhat limited in scope. I suggest expanding this section to include other factors that may affect study outcomes, such as varying definitions of adverse events or different follow-up periods across trials. A more thorough sensitivity analysis would improve the robustness of the findings.

Authors’ response: A more thorough sensitivity analysis was presented in the revised manuscript. 

Reviewer #2:

1. A major concern is the inclusion criteria. The authors intend to include previously untreated cervical cancer patients, however, currently, immunotherapy is usually used in late stage or recurrent cases as a salvage strategy. Only including previously untreated cases might appear somewhat biased and does not reflect actual clinical practice. I would suggest broadening the inclusion criteria to all cervical cancer patients received pembrolizumab treatment.

Other PD-1 inhibitors, such as Nivolumab should be included as well if there are available studies.

Authors’ response: We appreciate the recommendation and have changed the inclusion criteria to all cervical cancer patients who received pembrolizumab treatment. However, the systematic review will address the question of the effectiveness and safety of pembrolizumab, and therefore, it is not correct to include patients treated with Nivolumab. 

2. In search strategy, (lines #129-136) the authors included pembrolizumab and bevacizumab, but left out other standard treatment strategies, such as chemotherapy, radiotherapy, or surgery.

Authors’ response: We appreciate the suggestion for improving the search strategy and the suggested terms have been added accordingly. 

3. Line #152, I don’t think the patients have to be “recently” diagnosed and suggest “recently” be strike out.

Authors’ response: Thank you. The word “recently” has been removed as suggested.

4. Line #158, I think bevacizumab should be included as well.

Authors’ response: Thank you. Bevacizumab was included.

5. For the subgroup analysis, I suggest look at subgroups divided by expression levels of immune checkpoint molecules, such as PD-1, PD-L1, to see if certain molecular signatures are associated with ICI treatment effects. This will shed lights on what patient populations will benefit most from ICIs.

Authors’ response: We have included the analysis of subgroups divided by expression levels of immune checkpoint molecules, such as PD-1 and PD-L1. 

6. Line #184, “DMB” should be “DMD”.

Authors’ response: Thank you. Fixed as suggested.

7. Line #45, “will be included” should be deleted for a correct sentence structure.

Authors’ response: Thank you. Fixed as suggested.

Reviewer #3

1. Comprehensive Literature Search: Your protocol outlines a focused search strategy that, while robust, may benefit from an expanded scope to include grey literature and databases such as The World Health Organization (WHO) International Clinical Trials Registry Platform (ICTRP) search portal (http://apps.who.int/trialsearch); Clinical Trials Registry Platform (http://www.who.int/ictrp/en/); European Medicines Agency (EMA)(https://www.ema.europa.eu/ema/); US Food and Drug Administration (FDA) (www.fda.gov). To ensure even more comprehensive coverage, please consider meticulously scanning the abstracts presented at major international congresses during the three years preceding our search. This strategy aims to capture any studies that have been presented at these events but have yet to be fully published. This broader approach will help capture emerging studies and ensure a more comprehensive analysis, enhancing the relevance and applicability of your findings.

Authors’ response: We have included the suggested databases in our search strategy. In addition, we have included the search for abstracts published in international conferences in the last 5 years. 

2. Search Terms and Inclusion Criteria: The manuscript currently limits its search terms primarily to specific names associated with Bevacizumab. Considering the diversity of treatment modalities in cervical cancer, including chemotherapy, radiotherapy, and surgery, broadening the search terms to encompass these areas could provide a more holistic view of the landscape and potential interactions or comparative efficacies.

Authors’ response: We appreciate the suggestion for improving the search strategy and the suggested terms have been added. 

3. Discussion of Combined Therapies: Your manuscript would benefit from a more detailed discussion on the role and implications of combining Pembrolizumab with Bevacizumab. Highlighting this in both the introduction and discussion sections would clarify the therapeutic context and relevance of your findings to current treatment protocols.

Authors’ response: The role and implications of combining Pembrolizumab with Bevacizumab were properly added to the introduction and discussion sections.

4. Analysis of Adverse Events: Considering serious adverse events as a secondary endpoint is a prudent choice.

However, a comprehensive analysis of all adverse events, if feasible, would provide a deeper understanding of safety profiles, which is crucial for clinical decision-making.

Authors’ response: A detailed analysis of the outcomes, including adverse events, was provided in the GRADE table in the revised manuscript. 

5. Handling of Missing Data: It is essential to outline a clear and rigorous method for handling missing data in your analysis. This includes efforts to retrieve missing information and statistically estimating unavailable data, which would strengthen the validity and reliability of your study results.

Authors’ response: Throughout the revised text we have clarified how we will deal with missing data and even added a specific subsection on this. 

6. Clarity of Research Questions: Finally, I recommend structuring a separate section that distinctly lists the research questions your study seeks to answer, including benefits, risks, and subgroup analyses. This would greatly enhance the clarity and focus of your manuscript, making it more accessible to readers and reviewers.

Authors’ response: We added a subsection on the research question, where the systematic review question was placed. 

Reviewer #4: 

The author expressed the need for further research on pembrolizumab as a treatment option for cervical cancer.

However, the outcome selection seems to be not ideal.

Previous meta-analysis used the outcome of complete response (CR), partial response (PR), stable disease (SD), disease progression (PD), overall response rate (ORR), disease control rate (DCR), median progression-free survival (mPFS), and median overall survival (mOS)

The latest RCT reported OS and PFS.

Authors need to re-consider the outcome that would be included.

Authors’ response: We agree that the suggested outcomes are important and have added them all in the revised manuscript, including a definition for each one. 

Reviewer #5:

1. The research question is clearly stated and addresses a 

---

## [Decision Letter · Decision Letter 1]

30 Sep 2024

Efficacy and safety of Pembrolizumab in cervical cancer: protocol for systematic review and meta-analysis of randomized clinical trials

PONE-D-24-34168R1

Dear Dr. Ricardo Ney Oliveira Cobucci,

Ricardo Ney Oliveira CobucciWe’re pleased to inform you that your manuscript has been judged scientifically suitable for publication and will be formally accepted for publication once it meets all outstanding technical requirements.

Kind regards,

Cho-Hao Howard Lee, M.D.

Academic Editor

PLOS ONE

Reviewers' comments:

Reviewer's Responses to Questions

**Comments to the Author**

1. Does the manuscript provide a valid rationale for the proposed study, with clearly identified and justified research questions?

Reviewer #1: Yes

Reviewer #2: Yes

Reviewer #3: Yes

Reviewer #4: Yes

Reviewer #5: Yes

2. Is the protocol technically sound and planned in a manner that will lead to a meaningful outcome and allow testing the stated hypotheses?

Reviewer #1: Yes

Reviewer #2: Yes

Reviewer #3: Yes

Reviewer #4: Yes

Reviewer #5: Yes

3. Is the methodology feasible and described in sufficient detail to allow the work to be replicable?

Reviewer #1: Yes

Reviewer #2: Yes

Reviewer #3: Yes

Reviewer #4: Yes

Reviewer #5: Yes

4. Have the authors described where all data underlying the findings will be made available when the study is complete?

Reviewer #1: Yes

Reviewer #2: Yes

Reviewer #3: Yes

Reviewer #4: Yes

Reviewer #5: Yes

5. Is the manuscript presented in an intelligible fashion and written in standard English?

Reviewer #1: Yes

Reviewer #2: Yes

Reviewer #3: Yes

Reviewer #4: Yes

Reviewer #5: Yes

6. Review Comments to the Author

You may also provide optional suggestions and comments to authors that they might find helpful in planning their study.

Reviewer #1: Dear Authors,

Thank you for submitting the revised version of your manuscript titled "Efficacy and Safety of Pembrolizumab in Cervical Cancer: Protocol for Systematic Review and Meta-Analysis of Randomized Clinical Trials." After thoroughly reviewing your revisions, I am pleased to say that you have addressed all of the previous concerns and suggestions with a high degree of care and attention to detail. The manuscript now meets the necessary standards for publication.

Strengths:

Comprehensive Search Strategy: The detailed and robust search strategy, which now includes additional databases and a clear outline of grey literature handling, ensures that the systematic review will capture the most relevant studies on pembrolizumab in cervical cancer. Your methodical approach is well-structured and leaves little room for gaps in the literature.

Clear Subgroup and Sensitivity Analyses: The revisions provide a much clearer explanation of how subgroup and sensitivity analyses will address potential heterogeneity in treatment combinations and dosages. The inclusion of molecular signatures, such as PD-1 and PD-L1 expression, in subgroup analyses is particularly valuable and enhances the methodological rigor of the study.

Well-Defined Outcome Measures: I appreciate the additional clarity and justification regarding the primary and secondary outcomes. The inclusion of a broader range of safety outcomes in your adverse event analysis ensures a more comprehensive evaluation of pembrolizumab's safety profile, which is critical for informing clinical decision-making.

Thorough Risk of Bias and GRADE Assessments: Your attention to the risk of bias in grey literature, as well as the detailed explanation of how the GRADE framework will be applied, strengthens the transparency and reliability of the forthcoming systematic review. This will provide clinicians and researchers with high-quality evidence to guide treatment decisions.

Strong Statistical Plan: The updated statistical analysis plan, which now includes clearer details on how missing data will be handled and how fixed-effects or random-effects models will be applied, provides a solid foundation for the meta-analysis. This will ensure that the synthesis of evidence is both rigorous and reliable.

Conclusion:

In my view, the manuscript is now well-prepared for publication. The revisions have significantly improved the clarity, depth, and methodological rigor of the protocol. The systematic review and meta-analysis proposed in this study have the potential to offer meaningful insights into the efficacy and safety of pembrolizumab in the treatment of cervical cancer, which is a highly relevant topic in oncology today. I believe this work will be a valuable addition to the literature, and I recommend it for acceptance without further revision.

Reviewer #2: The authors have revised according to my suggestions and provided resonable justification for the parts that remain unchanged. In all, this protocol is ready for publication.

Reviewer #3: I agree with the author's revisions. However, I have one small comment: the newly added text in the Revised Manuscript (without Track Changes) has a gray background. This should be removed prior to publication.

Reviewer #4: No further recommendations. Hopefully this protocol and the following study will be able to address the efficacy of Pembrolizumab in cervical cancer.

Reviewer #5: I appreciate the revision from the authors and the manuscript now looks good to me. I wish the best of luck to their research endeavors.

7. PLOS authors have the option to publish the peer review history of their article (what does this mean?). If published, this will include your full peer review and any attached files.

Reviewer #1: **Yes: **Xiaoyi Zhang, MD

Reviewer #2: No

Reviewer #3: No

Reviewer #4: No

Reviewer #5: **Yes: **Yuhang Liu

---

## [Editor Report · Acceptance letter]

2 Oct 2024

PONE-D-24-34168R1 

PLOS ONE

Dear Dr. Cobucci, 

I'm pleased to inform you that your manuscript has been deemed suitable for publication in PLOS ONE. Congratulations! Your manuscript is now being handed over to our production team.

Kind regards, 

on behalf of

Dr. Cho-Hao Howard Lee 

Academic Editor

PLOS ONE